# Research on the Hot Deformation Behavior of the Casting NiTi Alloy

**DOI:** 10.3390/ma14206173

**Published:** 2021-10-18

**Authors:** Chengchuang Tao, Hongjun Huang, Ge Zhou, Bowen Zheng, Xiaojiao Zuo, Lijia Chen, Xiaoguang Yuan

**Affiliations:** School of Materials Science and Engineering, Shenyang University of Technology, Shenyang 110870, China; tao_chengc@163.com (C.T.); huanghong1977@163.com (H.H.); zhengbowen89@163.com (B.Z.); kz_candy@163.com (X.Z.); chenlijia@sut.edu.cn (L.C.); yuanxg@sut.edu.cn (X.Y.)

**Keywords:** casting NiTi alloy, activation energy, strain rate sensitivity exponent, constitutive equation, processing map, microstructure

## Abstract

The hot deformation behavior and processing maps of the casting NiTi alloy were studied at the deformation temperature of 650–1050 °C and the strain rate of 5 × 10^−3^–1 s^−1^ by Gleeble-3800 thermal simulating tester. The variation of the strain rate sensitivity exponent *m* and the activation energy *Q* under different deformation conditions (T = 650–1050 °C, $\dot{\varepsilon }$ = 0.005–1 s^−1^) were obtained. The formability of the NiTi alloy was the best from 800 °C to 950 °C. The constitutive equation of the casting NiTi alloy was constructed by the Arrhenius model. The processing map of the casting NiTi alloy was plotted according to the dynamic material model (DMM) based on the Prasad instability criterion. The optimal processing areas were at 800–950 °C and 0.005–0.05 s^−1^. The microstructure of the casting NiTi alloy was analyzed by TEM, SEM and EBSD. The softening mechanisms of the casting NiTi alloy were mainly dynamic recrystallization of the Ti_2_Ni phase and the nucleation and growth of fine martensite.

## 1. Introduction

The NiTi shape memory alloy was firstly discovered at the US Navy Ordnance Laboratory. The NiTi alloy possesses the advantage of shape memory effect (SME), super elastic (SE) and excellent corrosion resistance [1,2,3], which is widely used in automotive, aerospace, and other fields [4,5,6,7,8,9]. Chang et al. [10] found the optimal shape memory effect of the NiTi alloy was obtained at 300 °C × 100 h through studying the influence of aging process on the shape memory effect and superelastic properties. Fan et al. [11] investigated the phase transformation behavior of the Ni51.1-Ti (at.%) alloy by thermodynamic and experimental methods. It was found that the phase transformations from B2 phase to R phase and then from R phase to B’19 phase occurred during the cooling process after the aging treatment at 500–550 °C. Takeda et al. [12] researched the influence of creep conditions on the volume fraction of the martensitic phase of the NiTi alloy by studying the thermodynamic properties of the NiTi alloy. Therefore, the NiTi binary alloy has become one of the hottest topics.

Many scholars [13,14] have analyzed the mechanical properties and microstructure of the NiTi alloy in detail. Luo et al. [15,16] investigated the grain size, the recrystallization fraction, the orientation of the grain boundary and the evolution of second phase particles of the hot-extruded NiTi alloy by transmission electron microscopy (TEM) and electron backscatter diffraction (EBSD). Motemani et al. [17] revealed the effect of Ni_3_Ti on the transformation temperature of the martensitic by cooling the hot-rolled Ni50.7-Ti (at.%) alloy at different rates. Bhagyaraj et al. [18] found the influence of the second phase on the deformation of the 50.2Ti-Ni (at.%) alloy based on rolling experiments, and the evolution law of the second phase and microstructure during rolling was analyzed by SEM and TEM. Sun et al. [19] proposed the effect of different heat treatment parameters on the grain size and the mechanic property of the isoatomic NiTi alloy after compressing. However, most research focuses on the heat treatment processes, microstructures and properties of the NiTi alloy; the hot deformation mechanism of the casting NiTi alloy has attached little attention. Therefore, it is significantly important to research the high-temperature deformation mechanism of NiTi alloys, and determine the optimal hot processing parameter through establishing the constitutive equation, for the majorization of the hot deformation process such as rolling. Zhang et al. [20] conducted hot compression experiments on the NiTi alloy and a constitutive equation of the macroscopic stress was constructed by revising the friction and temperature during the experiment, which provided a model for numerical simulation of the high-temperature forming process of the NiTi alloy. Jiang et al. [21] investigated the effects of the strain rate, the deformation temperature, and the deformation degree on dynamic recovery and dynamic recrystallization of the NiTi alloy. Zhang et al. [22] identified that the instability zone of the hot-processing map was directly proportional to the degree of deformation through establishing the hot-processing map and the constitutive equation of the NiTi alloy, and the optimal processing temperature was 750–900 °C. Yang et al. [23] proposed the constitutive model $\sigma_{0}^{P}=1+K\mu e_{eq}^{P}$ through quantitatively decomposing the macro strain by neutron diffraction. Nayan et al. [24] studied the effect of Cu and Fe on the hot deformation mechanism of the hot-rolled NiTi alloy and constructed the hot-processing map. However, most of the above studies focus on the evolution of microstructure, the constitutive relationship and the deformation mechanism of the hot rolled NiTi alloy. Only a few researchers have studied the as-cast NiTi alloy. Therefore, this paper aims to study the hot deformation behavior and the deformation mechanism of the as-cast NiTi alloy. The microstructure evolution of the NiTi alloy was studied by SEM, TEM and EBSD, which provides a theoretical basis for the NiTi alloy in hot process.

## 2. Experimental

The experimental material is the NiTi alloy, which is smelted by vacuum consumable smelting furnace. The chemical composition of the NiTi alloy is listed in Table 1. The NiTi alloy was cut into the sample with a size of φ8 × 12 mm, and the oxide scale of the sample was polished off. The high-temperature compression test was carried out on the Gleeble-3800 thermal-mechanical simulation tester. The deformation temperature was from 650 °C to 1050 °C, the strain rate was from 5 × 10^−3^ s^−1^ to 1 s^−1^, and the strain was 0.7. The variation rate of the temperature was 10 °C/s. The test was conducted under low vacuum. The friction was reduced through using Graphite lubrication. In order to make the experimental results accurate, 20 groups of samples were used in different parameters [20,21,24]. The complete high-temperature microstructure of NiTi alloy after compression was obtained by the water cooling, as shown in Figure 1.

The sample was selected in the axial direction. The samples were analyzed by Shimadzu 7000 X-ray diffraction (XRD, Shimadzu, Kyoto, Japan). The scanning angle was from 20° to 90°and the scanning speed was 8°/min. The equipment type of EBSD was GeminiSEM-300 (Carl Zeiss, Jena, Germany). The experimental step was 2 μm and the voltage was 20 kV. The samples were observed in JEM-2100 transmission electron microscope (TEM, JEOL, Tokyo, Japan) and Hitachi S-3400N scanning electron microscope (SEM, Hitachi, Kyoto, Japan). The constituent of the corrosive medium was H_2_O:HNO_3_:HF = 10:2:1. Moreover, the TEM sample was prepared by the Tenupol-5 double jet equipment (Struers, Shanghai China), and the reagent was CH_3_OH:HNO_3_ = 3:1. The experiment was conducted at a potential difference of 20 V and −20 °C.

## 3. Results and Discussion

### 3.1. Mechanical Characteristics of the Casting NiTi Alloy during High-Temperature Deformation

The high temperature compression test is conducted under different deformation parameters (T: 650–1050 °C, $\dot{\varepsilon }$: 0.005–1 s^−1^), and the stress vs. strain curves of the NiTi alloy after the high-temperature compression test are established, as exhibition in Figure 2. With the increase of the strain, the flow curve includes three stages, which are work hardening, softening and steady state, and the process is usually formed by the interaction of work hardening and softening. At the initial stage of compression, the number of dislocations increases, and the hardening mechanism is stronger than the softening mechanism due to the accumulation of high-density dislocations. With the increase of strain, the stress decreases, which is attributed to that the softening effect is higher than the work hardening. After that, the stress reaches the stable stage, and the effect of the hardening and the softening are in equilibrium. In Figure 2, the flow stress and its rate change increase with the increase of the strain rate, which indicates that the NiTi alloy is sensitive to the rate change of the strain at the same temperature. In addition, with the increase of deformation temperature and the decrease of the strain rate, the peak-stress of the NiTi alloy decreases, as plotted in Figure 2f.

### 3.2. Strain Rate Sensitivity Exponent of the NiTi Alloy

The essential characteristics of the hot deformation process of metallic materials conform to the Backofen [25] Equation (1),
(1)$$\sigma =K{\dot{\varepsilon }}^{m}$$
where, *σ* is the flow stress (MPa); $\dot{\varepsilon }$ is the strain rate (s^−1^); K is a constant; *m* is the sensitivity exponent of the strain rate. Equation (2) can be obtained by the mathematical transformation from Equation (1).
(2)$$m=\frac{d\ln\sigma }{d\ln\dot{\varepsilon }}$$

The sensitivity exponent of the strain rate is an important parameter which reflects the thermoforming property of the alloy. The larger value of *m* is, the better the formability of the alloy is. From the discussion in Section 3.1, the NiTi alloy belongs to materials that are sensitive to the strain rate. Figure 3a shows the variation of the *m* value at 650 °C, 750 °C, 850 °C, 950 °C and 1050 °C with the strain of 0.4. The slope of the line represents the *m* value of the NiTi alloy. Figure 3b shows the *m* values of the deformation with the strain of 0.1–0.6 at different temperatures, as plotted in Table 2.

Figure 3 and Table 2 show that the *m* value of NiTi alloy keeps the same trend under different degrees of deformation. The *m* firstly increases and then decreases with the increase of the deformation temperature, and reaches the maximum value of 0.227 at 950 °C. When the temperature increases to 1050 °C, the *m* value decreases, which is due to the change of the microstructure of the NiTi alloy during high-temperature deformation.

### 3.3. Hot Deformation Activation Energy of the Casting NiTi Alloy

High-temperature deformation of the NiTi alloy is a hot activation process based on the energy dissipation and redistribution. The influence of parameters such as temperature, the strain rate and the degree of the deformation on the flow stress can be described by the Arrhenius hyperbolic sine function model [26,27], as shown in Equation (3). Under low stress level (*ασ* ≤ 0.8), sinh(*ασ*) is *ασ*, and under high stress level (*ασ* ≥ 1.6), sinh(*ασ*) is 0.5exp(*ασ*). Equations (4) and (5) are transformed form Equation (3).
(3)$$\dot{\varepsilon }=A{\left[\sinh(\alpha \sigma )\right]}^{n}\exp\left(-\frac{Q}{RT}\right)\;$$
(4)$$\dot{\varepsilon }=A_{1}\sigma^{n_{1}}\exp\left(-\frac{Q}{RT}\right)\;\left(\alpha \sigma \leq 0.8\right)$$
(5)$$\dot{\varepsilon }=A_{2}\exp\left(\beta \sigma \right)\exp\left(-\frac{Q}{RT}\right)\;\left(\alpha \sigma \geq 1.6\right)$$
where, *A*, *A*_1_, *A*_2_, *α*, *β* are the constant, $\dot{\varepsilon }$ is the strain rate (s^−1^), *σ* is the flow stress (MPa), *n* and *n*_1_ are the stress exponents, *Q* is the activation energy (KJ/mol), *R* is the gas constant (J/(mol·K)), *T* is the deformation temperature (K). In Equations (4) and (5), *A*_1_ equals to *Aα^n^*_1_, *A*_2_ equals to *A*/2*^n^* and *β* equals to *αn*_1_. Equations (6) and (7) are obtained by the logarithmic transformation from Equations (4) and (5) respectively.
(6)$$\ln\dot{\varepsilon }=\ln A_{1}+n_{1}\ln\sigma -\frac{Q}{RT}$$
(7)$$\ln\dot{\varepsilon }=\ln A_{2}+\beta \sigma \;\;-\frac{Q}{RT}$$

The values of *n*_1_, *β* and *α* are calculated from Equations (6) and (7) by fitting the relationship curves of ln$\dot{\varepsilon }$ vs. ln*σ* and ln$\dot{\varepsilon }$ vs. *σ*, as shown in Figure 4. Relevant parameters of the NiTi alloy under different strain are calculated and listed in Table 3.

Equation (8) can be obtained by the logarithmic transformation from Equation (3). The *n* can be acquired by fitting the curve of ln$\dot{\varepsilon }$ vs. ln[sinh(*ασ*)], as plotted in Figure 5. Equation (9) can be obtained through mathematical transformation from Equation (8). The activation energy *Q* of the NiTi alloy can be calculated by fitting the curve of ln[sinh(ασ)] vs. T^−1^ and the *n* value.
(8)$$\ln\dot{\varepsilon }=\ln A+n\ln\left[\sinh\left(\alpha \sigma \right)\right]-\frac{Q}{RT}$$
(9)$$Q=nR{\left[\frac{\partial \ln\left[\sinh\left(\alpha \sigma \right)\right]}{\partial T^{-1}}\right]}_{\dot{\varepsilon }}$$

The activation energy *Q* at the strain of 0.4 is calculated by Equation (9), and the calculation result is listed in Table 4.

The activation energy *Q* represents the energy dissipation and redistribution of the material during the deformation process. Therefore, the *Q* value can truly reflect the change of the energy of the NiTi alloy. Figure 6 illustrates the relationship between the *m* value and the *Q* value of the NiTi alloy at the strain of 0.4, and it can be seen from Table 4 and Figure 6 that the *Q* value ranges from 220.98 KJ/mol to 439.31 KJ/mol. The inflection point of the *Q* value is 227.37 KJ/mol which appears at the deformation temperature of 850 °C. This phenomenon is mainly caused by the acute movement of the dislocation in the alloy with the increase of temperature, which is conducive to the formation of sub grain boundary and the large angle grain boundary. During the deformation process, the migration of large angle grain boundaries produces recrystallization, which is beneficial to the deformation and leads to the decrease of *Q* value. Based on the above analysis, the best formability of the NiTi alloy is obtained at 800–950 °C.

### 3.4. Hot Deformation Constitutive Relation of the Casting NiTi Alloy

The Arrhenius-type constitutive equation [28,29] is used to describe the deformation process of the alloy, which can better predict the ability of the hot deformation and the microstructure evolution of the NiTi alloy. The α, n and Q are calculated through bringing the peak-stress into Equations (6) and (9), which is 0.00928, 3.7168 and 272.69 KJ/mol respectively. The Q values of the Ni-50 wt.%Ti alloy and the Ni-44 wt.%Ti alloy are 304.41 KJ/mol and 267.89 KJ/mol calculated by Zhang [22] and Wang [30], which prove that the Q calculated in this article is reliable.

Zener-Hollomon parameter [30] can be used to characterize the relationship between the temperature and the strain rate, as shown in Equation (10). Equation (11) is obtained by the logarithmic calculation from Equation (10). Figure 7 illustrates the relationship between lnZ and lnsinh(*ασ*) at different temperatures and strain rates. Based on linear fitting analysis, the relationship between lnZ and lnsinh(*ασ*) of the NiTi alloy can be expressed by Equation (12), and the value of *A* is calculated as 1.556 × 10^11^.
(10)$$Z=\dot{\varepsilon }\exp\left(\frac{Q}{RT}\right)=A{\left[\sinh\left(\alpha \sigma \right)\right]}^{n}$$
(11)lnZ=lnA+nlnsinh(ασ)
(12)lnZ=25.771+3.63745lnsinh(ασ)

The *Q*, *A*, *α*, *n* are brought into Equation (3) and the constitutive relationship of the NiTi alloy can be obtained as follows:(13)$$\dot{\varepsilon }=1.556\times 10^{11}{\left[\sinh(0.00928\sigma )\right]}^{3.7168}\exp\left(-\frac{2.7269\times 10^{5}}{RT}\right)\;$$

Equation (14) can be derived from Equation (10). Equation (15) can be obtained by definition of inverse hyperbolic sine function. The constitutive equation of the NiTi alloy is expressed by Z parameters by plugging the material parameters into Equation (16).
(14)$$\sigma =\frac{1}{\alpha }\arcsin h\left[{\left(\frac{Z}{A}\right)}^{\frac{1}{n}}\right]$$
(15)$$\sigma =\frac{1}{\alpha }\ln\left\{{\left(\frac{Z}{A}\right)}^{\frac{1}{n}}+{\left[{\left(\frac{Z}{A}\right)}^{\frac{2}{n}}+1\right]}^{\frac{1}{2}}\right\}$$
(16)$$\sigma =\frac{1}{0.00928}\ln\left\{{\left(\frac{Z}{1.556\times 10^{11}}\right)}^{\frac{1}{3.7168}}+{\left[{\left(\frac{Z}{1.556\times 10^{11}}\right)}^{\frac{2}{3.7168}}+1\right]}^{\frac{1}{2}}\right\}$$

The fitting curve between the experimental peak of the stress and the calculated one is plotted in Figure 8. The linear regression correlation coefficient R^2^ is 0.992, which indicates the high accuracy of the constitutive relationship of the casting NiTi alloy.

### 3.5. Hot-Processing Map Construction of the Casting NiTi Alloy

Prasad et al. [31] firstly proposed the material dynamic model (DMM) based on mechanics, physical simulation and thermodynamics, and then Rao and Murty [32,33] improved the DMM. Microstructural evolution of materials during the hot deformation is analyzed by DMM in terms of the energy conversion. The DMM model can predict the processing property of materials and provide fundamental basis for the hot processing, which has been widely used in various alloys [34,35,36]. According to the DMM theory, the energy conversion in the hot deformation process of materials is composed of two parts, the dissipation quantity (*G*) and the dissipation coefficient (*J*). The relationship is as follows:(17)$$P=\sigma \varepsilon =G+J=\int_{0}^{\varepsilon }\sigma d\varepsilon +\int_{0}^{\sigma }\varepsilon d\sigma$$

The dissipation factor *η* is introduced to describe the utilization rate of energy consumed during the microstructure evolution of materials. *η* is defined as Equation (18). The criterion of the flow instability proposed by Prasad is used to determine the material instability zone, as exhibited in Equation (19).
(18)$$\eta =\frac{J}{J_{\max}}=\frac{2m}{m+1}$$
(19)$$\xi \left(\dot{\varepsilon }\right)=\frac{\partial \ln[m/(m+1)]}{\partial \ln\dot{\varepsilon }}+m<0\;$$

According to the above theory, the values of power dissipation factor *η* and flow instability criterion ξ are obtained. The hot-processing map of the NiTi alloy is plotted by combining power dissipation maps and instability maps, as illustrated in Figure 9. The hot-processing map can optimize the processing technology of materials and improve the workability of materials. Hot-processing maps can also be used to control the microstructure of materials, study the deformation mechanism, and analyze the instability region. In the hot-processing map of the NiTi alloy, the contour represents the power consumption coefficient, and the shadow part is the instability region. Figure 9 plots the instability region of the NiTi alloy, which increases with the increase of the strain. This is consistent with the existed conclusion [26,30,37]. When the strain is 0.2, the instability region of the alloy is mainly at 650 °C and 1050 °C under the high strain rate (1 s^−1^). With the increase of the strain, the instability region of the NiTi alloy gradually expands. When the strain is 0.6, the instability region of the alloy appears at 650 °C, 1 s^−1^ and 1000–1050 °C, 0.005–1 s^−1^, which is due to the deformation capacity of the alloy gradually decreasing with the increase of the strain at high temperature. In the stability region, the power dissipation η increases as the temperature increases or the strain rate decreases. The processing map shows that the formation process of NiTi is difficult at high strain rate (≥1 s^−1^), and the excellent forming performance is found at 800–950 °C. Figure 10 is the cracking diagram of hot deforming under different processing parameters, which illustrates that instability cracking occurs at 1050 °C, 0.005–1 s^−1^ and 650 °C, 1 s^−1^. At the deformation condition of 850–950 °C and 0.005–0.5 s^−1^, the smooth round cake-shape is specimen in the specimen after compression, and it can be concluded that the forming ability of the NiTi alloy at this condition is the best. Finally, the unstable region and the optimal processing region in the processing map are consistent with the experimental results.

### 3.6. Microstructure Analysis of the Casting NiTi Alloy

Figure 11 shows XRD analysis of the pre-and post-compression specimen and energy dispersive spectrometer (EDS) analysis of the as-cast NiTi alloy. Figure 11a illustrates that the composition of the pre-and post-compression specimen is mainly NiTi and Ti_2_Ni phase, and a small amount of Ni_3_Ti phase. The NiTi matrix phase and large Ti_2_Ni phase are found in the casting NiTi alloy at point A and B by EDS analysis, as displayed in Figure 11b.

Figure 12 shows TEM analysis of Ti_2_Ni phase in the NiTi alloy at 650 °C and the strain rate of 0.005 s^−1^. Based on the analysis of the diffraction spot, it can be concluded that the precipitation phase in the NiTi alloy is mainly Ti_2_Ni phase with the crystal band axis [$\bar{1}$14] which is consistent with the XRD and EDS results. Thus, the existence of the Ti_2_Ni phase in the NiTi alloy is further confirmed. Furthermore, it can be clearly seen from the diagram that the compressed strip-like Ti_2_Ni phase is in a larger size, which is about 11 μm. A smooth spreading surface is found on 1–1′ and 2–2′ area of the Ti_2_Ni phase, as shown in Figure 12. These expansion planes are low-index crystal planes, which contribute to the nucleation of the martensitic phase [18,38]. The microstructure characteristics around Ti_2_Ni phase can be observed from the extended surface of the Ti_2_Ni phase, and the extended surface can provide nucleation sites for the nucleation and growth of the martensite phase.

Figure 13 shows the SEM and TEM images of the NiTi alloy under different thermal deformation parameters. It can be seen that the large-scale Ti_2_Ni phase presents random distribution in the casting alloy. The EBSD analysis shows that the Ti_2_Ni phase is polycrystalline structure in Figure 13a. The microstructure of the alloy extends obviously along the compression direction, and the morphology and size of the Ti_2_Ni phase clearly change. At 850 °C and 0.005 s^−1^, the strip-like Ti_2_Ni phase transforms into recrystallized structure, and the recrystallized grains nucleate and grow along the original broken grain boundary. In addition, the recrystallized grains are equiaxed which have large size and huge quantity, which is due to the acute movement of the dislocation, and the plenty of dislocation movements induce the formation of subgrains at 850 °C, as presented in Figure 13b. The large angle grain boundaries are formed by sub grains merging and absorbing dislocations, and the nucleation of dynamic recrystallization is generated by migration. With the progress of deformation, the crystal nucleus grows into the recrystallized structure. Figure 13c illustrates the hot compression temperature further expands the instability area of the NiTi alloy. Figure 13d is TEM analysis of the alloy at 850 °C. The Ti_2_Ni phase and NiTi phase are calibrated by the diffraction spot. The dynamic recrystallization microstructure of the Ti_2_Ni phase can be clearly observed, and the fine martensite NiTi phase formed around the Ti_2_Ni phase. To conclude, the softening mechanisms of the NiTi alloy during the hot compression are a dynamic recrystallization of the Ti_2_Ni phase and the nucleation and growth of the fine martensite.

## 4. Conclusions

(1) The strain rate sensitivity index (*m*), activation energy (*Q*) and constitutive relationship are analyzed by studying the hot deformation behavior of the casting NiTi alloy. The *m* value of the alloy is 0.098–0.227, and the *Q* value is 220.98–439.31 kJ/mol. The constitutive relationship of the casting NiTi alloy is as Equation (13).

(2) The processing map of the casting NiTi alloy for different strains of 0.2, 0.4 and 0.6 is plotted according to the Prasad instability criterion. The instability region of the casting NiTi alloy expands with the increase of the strain. The optimal processing zone of the casting NiTi alloy is 800–950 °C, 0.005–0.5 s^−^^1^, and the instability zone is 650 °C, 1 s^−^^1^ and 1000–1050 °C, 0.005–1 s^−^^1^.

(3) During the hot deformation, the Ti_2_Ni phase provides nucleation sites for the martensitic phase in preference, and the recrystallization of Ti_2_Ni phase increases with the increase of the degree of the deformation and temperature. The softening mechanisms of the casting NiTi alloy during the hot compression are the dynamic recrystallization of the Ti_2_Ni phase and the nucleation and growth of fine martensite.

## Figures and Tables

**Figure 1 materials-14-06173-f001:**
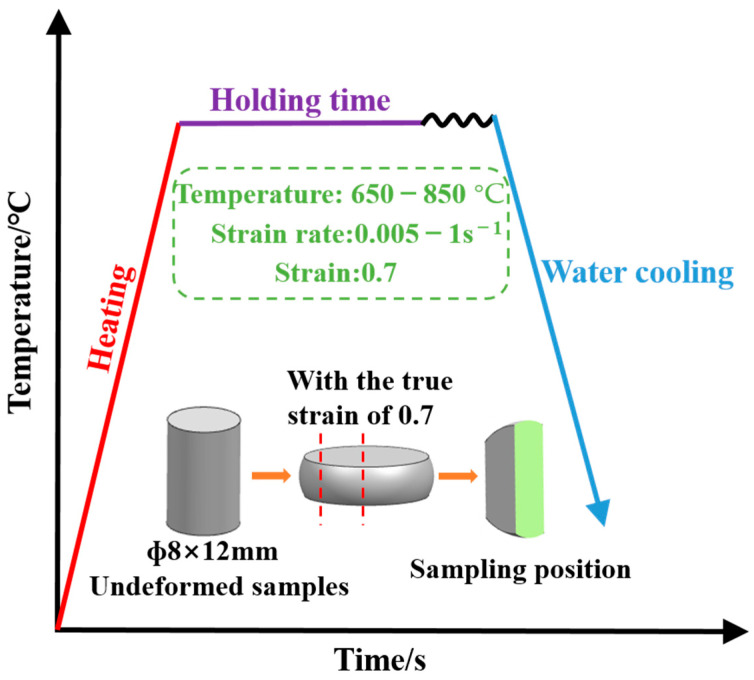
Experimental technique drawing.

**Figure 2 materials-14-06173-f002:**
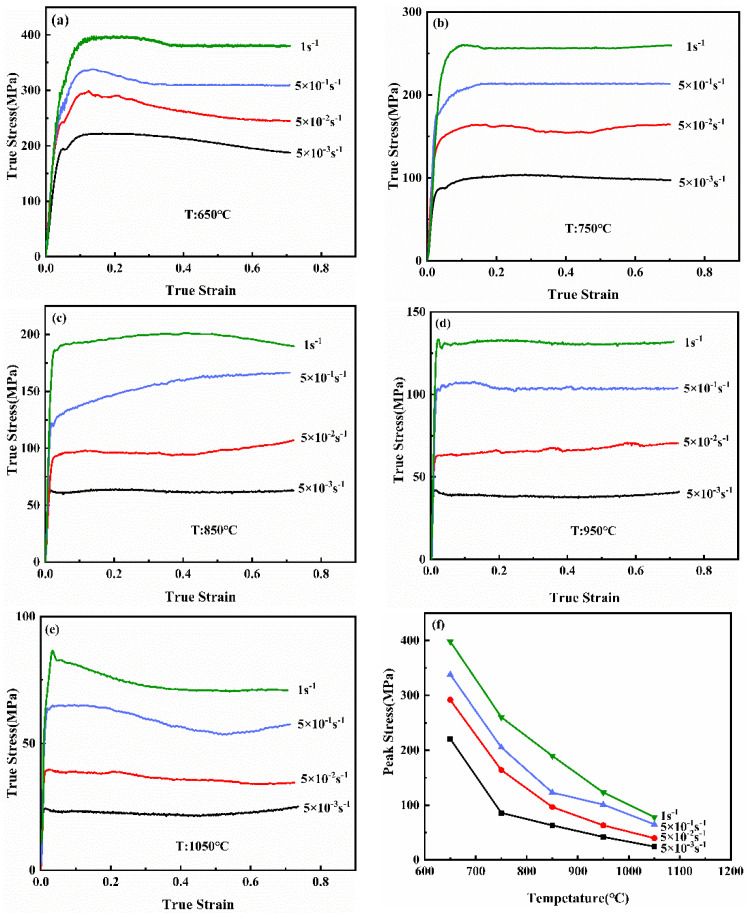
True stress-strain curves of the NiTi alloy at different deformation conditions. (**a**) 650 °C; (**b**) 750 °C; (**c**) 850 °C; (**d**) 950 °C; (**e**) 1050 °C; (**f**) Peak stress.

**Figure 3 materials-14-06173-f003:**
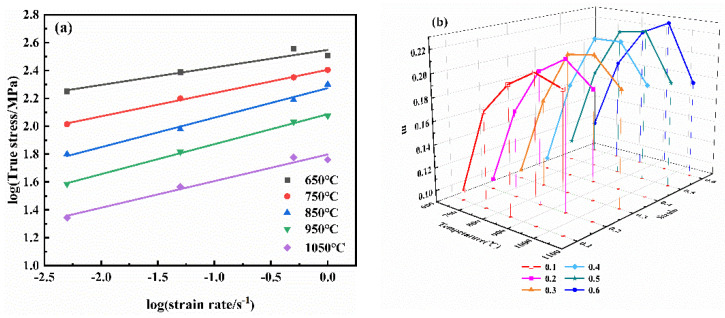
(**a**) log*σ*-log$\dot{\varepsilon }$ curves of NiTi alloy curves of the NiTi alloy, (**b**) Strain rate sensitivity *m* of the NiTi alloy at different temperatures.

**Figure 4 materials-14-06173-f004:**
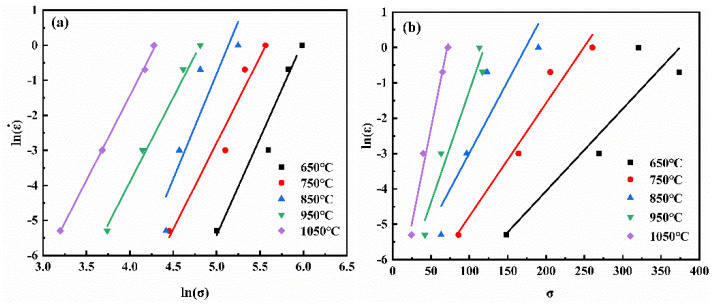
Fitting the relationship curves of ln$\dot{\varepsilon }$ vs. lnσ and ln$\dot{\varepsilon }$ vs. σ (ε = 0.4) (**a**) n_1_ value, (**b**) β value.

**Figure 5 materials-14-06173-f005:**
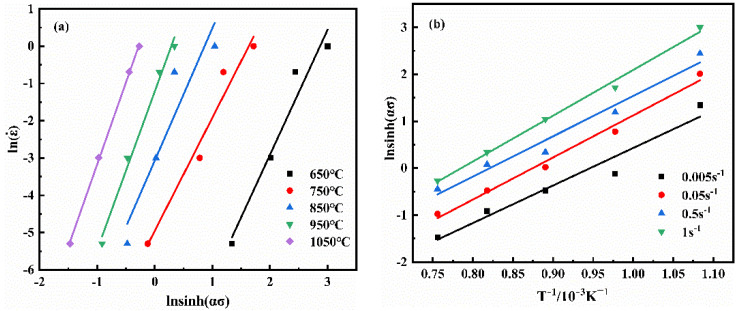
(**a**) ln($\dot{\varepsilon }$)-lnsinh(*ασ*) curves; (**b**) lnsinh(*ασ*)-T^−1^/10^−3^K^−1^curves of NiTi alloy (*ε* = 0.4).

**Figure 6 materials-14-06173-f006:**
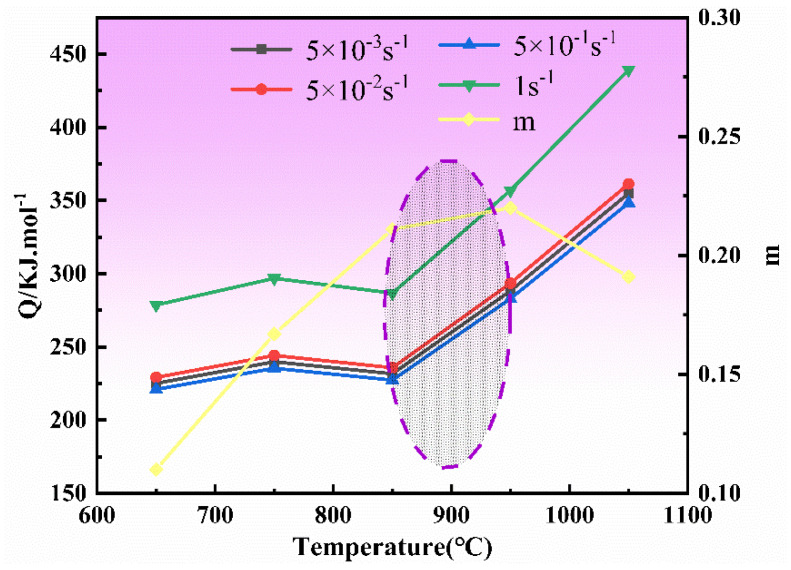
*Q*, *m* and T curves of NiTi alloy.

**Figure 7 materials-14-06173-f007:**
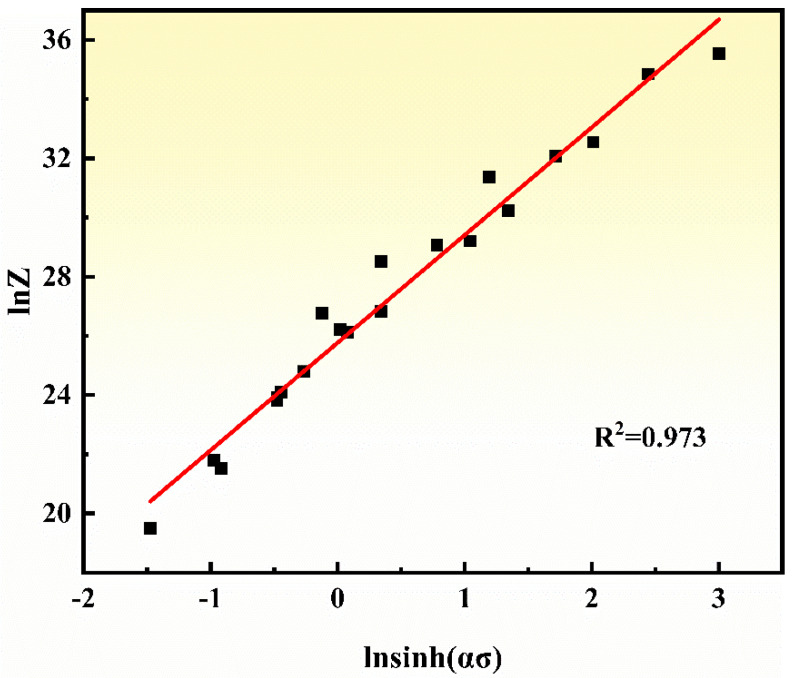
ln*Z*-lnsinh (*ασ*) curves of NiTi alloy.

**Figure 8 materials-14-06173-f008:**
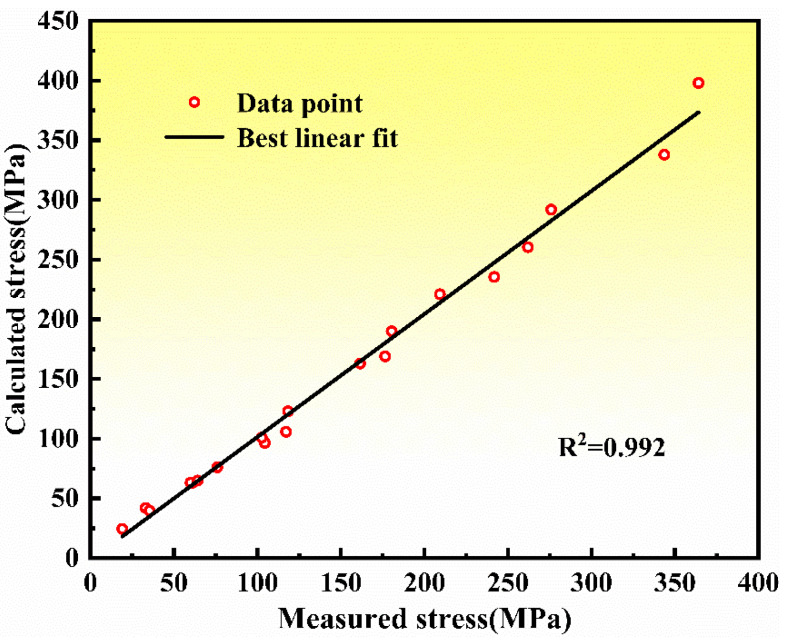
Calculated stress-measured stress fitting curves of the NiTi alloy.

**Figure 9 materials-14-06173-f009:**
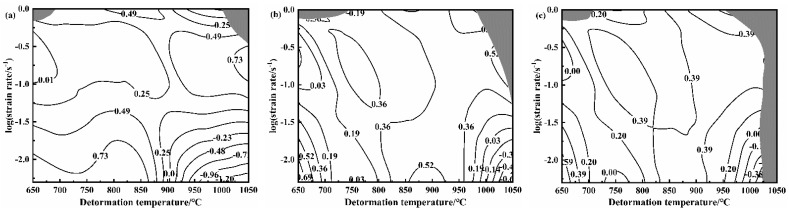
The processing maps for the NiTi alloy (**a**) ε = 0.2; (**b**) ε = 0.4; (**c**) ε = 0.6.

**Figure 10 materials-14-06173-f010:**
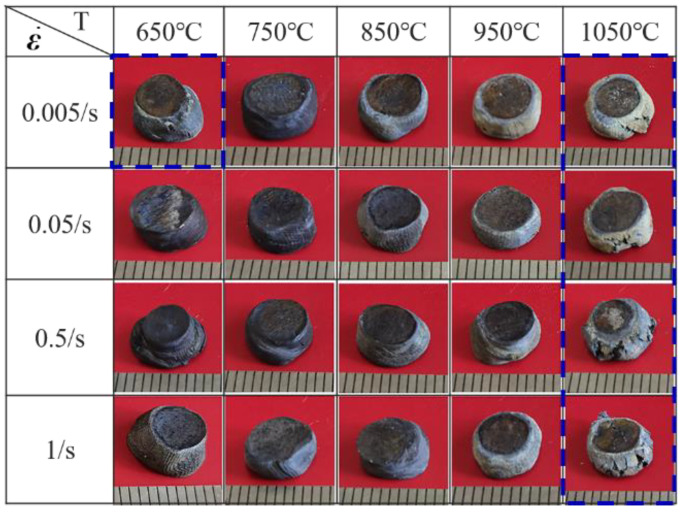
Cracking of NiTi alloy under different hot deformation processing parameters.

**Figure 11 materials-14-06173-f011:**
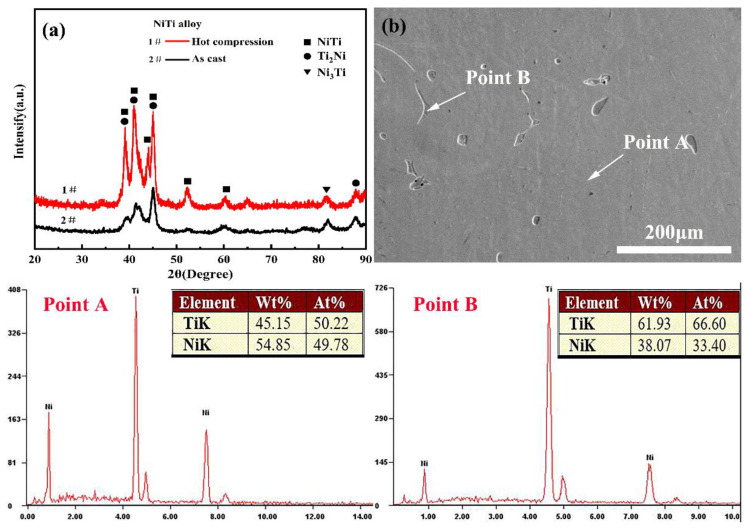
XRD and EDS analysis of NiTi alloy (**a**) XRD analysis; (**b**) EDS analysis.

**Figure 12 materials-14-06173-f012:**
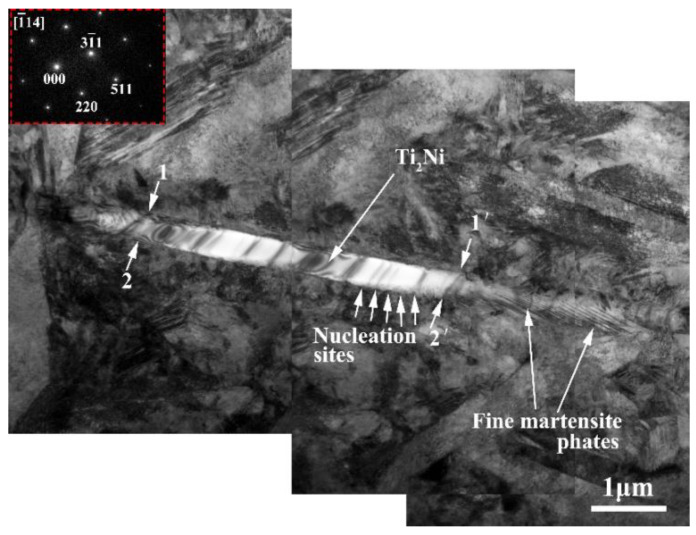
TEM analysis of Ti_2_Ni phase in NiTi alloy (T: 650 °C, $\dot{\varepsilon }$: 0.005 s^−1^).

**Figure 13 materials-14-06173-f013:**
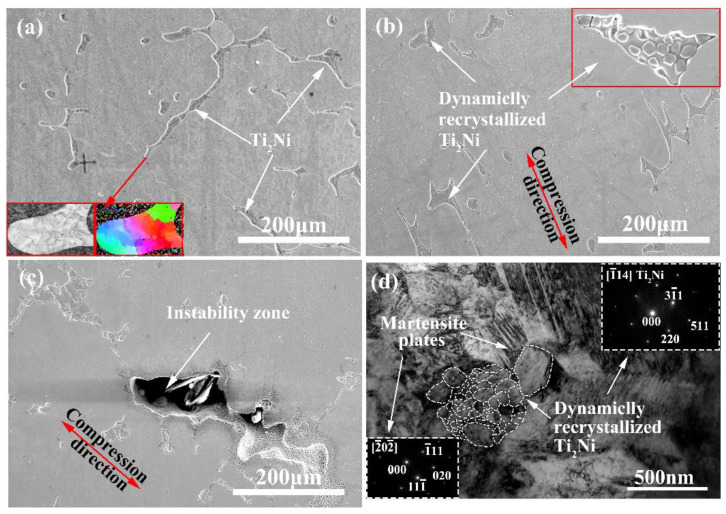
Microstructure analysis of the NiTi alloy at different temperatures with strain rate of 0.005 s^−1^ (**a**) as cast; (**b**) 850 °C; (**c**) 1050 °C; (**d**) TEM analysis at 850 °C.

**Table 1 materials-14-06173-t001:** The chemical composition of the NiTi alloy.

Element	Ni	Ti	Zr	Hf
Content (wt.%)	55.72	43.9	0.021	0.022

**Table 2 materials-14-06173-t002:** Strain rate sensitivity (*m*) of the NiTi alloy at different temperatures.

	T	650 °C	750 °C	850 °C	950 °C	1050 °C
ε						
0.1		0.098	0.172	0.199	0.213	0.205
0.2		0.099	0.165	0.203	0.217	0.198
0.3		0.09900	0.167	0.211	0.215	0.191
0.4		0.100	0.174	0.219	0.221	0.188
0.5		0.108	0.179	0.221	0.225	0.184
0.6		0.118	0.182	0.215	0.227	0.177

**Table 3 materials-14-06173-t003:** Values of *n*_1_, *β*, *α*, and *n* of NiTi alloy during hot deformation (*ε* = 0.1–0.6).

Strain	*n* _1_	*β*	*α*	*n*
0.1	5.93554318	0.00958788	0.05690933	3.72177449
0.2	5.97514982	0.00977601	0.05841309	3.69542632
0.3	5.97581742	0.01008106	0.06024256	3.65501741
0.4	5.89394579	0.01037857	0.06117069	3.58317916
0.5	5.74007542	0.01074333	0.06166753	3.48750191
0.6	5.63761810	0.01082401	0.06102161	3.47574367

**Table 4 materials-14-06173-t004:** Values of activation energies *Q* (KJ/mol) of NiTi alloy during hot deformation (*ε* = 0.4).

	T	650 °C	750 °C	850 °C	950 °C	1050 °C
*Ε*						
0.005		225.16	239.87	231.67	288.36	354.92
0.05		229.22	244.19	235.84	293.55	361.31
0.5		220.98	235.42	227.37	283.01	348.33
1		278.70	296.90	286.75	356.92	439.31

## Data Availability

The data presented in this study are available on request from the corresponding author.

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
