# Peer review of "Research on the Hot Deformation Behavior of the Casting NiTi Alloy"

_materials, 2021, doi:10.3390/ma14206173_

Round 1

Reviewer 1 Report

As-cast NiTi alloy was extensively studied. Several articles are reporting the same and using similar equations and methods. I can cite, for example, DOI: 10.1016/S1003-6326(16)64283-8, 10.1016/j.materresbull.2014.04.049, 10.1016/j.matdes.2010.05.048, 10.1007/s12598-019-01291-6, 10.1016/S1003-6326(11)61332-0. This set of articles (there are more) are better presented and give sound conclusions. 

The English of this article is very poor.

The article seems like a technical report.

There are missing references for some equations in sections 3.3 and 3.4, and the solutions of equations are either badly explained or meaningless.

The methodology to solve the Garofalo equation (Arrhenius type constitutive equation) is completely wrong.

Processing maps using only one strain is useless and leads one to wrong conclusions.

The phase Ti2Ni suddenly appeared; there was no chemical or XRD analysis to conclude its presence.

TEM analysis is poor. There is no diffraction analysis. Indeed, there is one diffraction pattern in Figure 10, but without any explanation.

Figure 12 is useless. The authors have drawn grains there and, again, did not present any evidence about the phase.

For sure, conclusions are not coherent.

Reviewer 2 Report

The authors studied the hot deformation behavior and processing maps of cast NiTi alloy, over the temperatures ranging from 650℃ to 1050℃. The microstructure of casting NiTi alloy was analyzed by TEM, SEM and EBSD.

Comments and Suggestions for Authors

Introduction: The introduction is well organized.

Experimental: Change the section Experimental with Materials and Methods.

Specify the SEM-EBSD analysis equipment and conditions.

How many samples were used? Please clearly add references of the study that you considered for the methodology.

Results and Discussion: Please provide more references. The discussion section must discuss the results obtained with the current literature.

Conclusions: The conclusions are supported by the results.

Bibliography: is consistent. She containing cites works representative for the approached domain.

Reviewer 3 Report

This manuscript provides an interesting study about Hot Deformation Behavior of Casting NiTi Alloy using Gleeble 3800 thermal-mechanical simulation test machine. The whole manuscript is well organized. There are a few comments for the authors to consider:

  1. The introduction session is too long. I would advice to break the whole paragraph into at least 2 paragraphs.
  2. 2 (a-e): the 5 figures were not arranged in an order way, but quite random way with respect to its temperature. Pls correct.
  3. Equation 2, 4: lg -->log?
  4. Line 155: N --> n?
  5. Line 156 and Table 2: Unit of activation energy Q , kJ/mol, should be given (instead of doing so at line 191). Units for other variables in all the equations should be provided.
  6. Equation 6 & 7: what do A1 and A2 represent?
  7. Line 210: material dynamic model (DMM) should be “material dynamic model (MDM)” or “dynamic material model (DMM)”?
  8. It is worthwhile to check through the English carefully. A few examples are below:
    1. Line 89: “water was cooled quickly to retain the complete high-temperature structure” --> “water cooling was quickly introduced to retain the complete high-temperature structure”
    2. Line 99: “and observed on the JEM-2100 99 transmission electron microscope.”--> “and the samples were observed in the JEM-2100 99 transmission electron microscope.”

The full scope of this manuscript is beyond my expertise, although I am very familiar with tensile test, TEM and EBSD used in this work to characterize metals

I went through the comments given by the 1st reviewer, and dig out the reference he/she mentioned. And indeed, a lot of more comprehensive researcher has been done about the hot deformation of NiTi shape memory alloy described in this manuscript. One example is attached.

As such, I would like to recommend rejecting this manuscript based on the following evidence:

This manuscript provides a study about Hot Deformation Behavior of Casting NiTi Alloy using Gleeble 3800 thermal-mechanical simulation test machine. The process map of NiTi has been studied intensively previously with the reference below:

  1. https://www.sciencedirect.com/science/article/abs/pii/S1003632616642838
  2. https://link.springer.com/article/10.1007/s11665-020-05182-1
  3. https://www.mdpi.com/2075-4701/7/9/328

Hence, this manuscript did not offer new information to the research community. The process map should be performed with a few strain rates, instead of a single value in this work. In addition, the labelling of equation 2 & 4 is not accurate (lg should be log). There is no clear definition of A1 and A2 in equation 6 & 7. 

Round 2

Reviewer 1 Report

The article has been improved. In the way it is now, I believe that it can be published as it is.